# RNAi-Based Biofungicides as a Promising Next-Generation Strategy for Controlling Devastating Gray Mold Diseases

**DOI:** 10.3390/ijms21062072

**Published:** 2020-03-18

**Authors:** Md Tabibul Islam, Sherif M. Sherif

**Affiliations:** Alson H. Smith Jr. Agricultural Research and Extension Center, School of Plant and Environmental Sciences, Virginia Tech, Winchester, VA 22602, USA; tabibul@vt.edu

**Keywords:** *Botrytis cinerea*, spray-induced gene silencing, RNAi-based biofungicide, siRNA, dsRNA

## Abstract

*Botrytis cinerea* is one of the most critical agro-economic phytopathogens and has been reported to cause gray mold disease in more than 1000 plant species. Meanwhile, small interfering RNA (siRNA), which induce RNA interference (RNAi), are involved in both host immunity and pathogen virulence. *B. cinerea* has been reported to use both siRNA effectors and host RNAi machinery to facilitate the progression of gray mold in host species. Accordingly, RNAi-based biofungicides that use double-stranded RNA (dsRNA) to target essential fungal genes are considered an emerging approach for controlling devastating gray mold diseases. Furthermore, spray-induced gene silencing (SIGS), in which the foliar application of dsRNA is used to silence the pathogen virulence genes, holds great potential as an alternative to host-induced gene silencing (HIGS). Recently, SIGS approaches have attracted research interest, owing to their ability to mitigate both pre- and post-harvest *B. cinerea* infections. The RNAi-mediated regulation of host immunity and susceptibility in *B. cinerea*–host interactions are summarized in this review, along with the limitations of the current knowledge of RNAi-based biofungicides, especially regarding SIGS approaches for controlling gray mold diseases under field conditions.

## 1. Introduction

*Botrytis cinerea* has been reported to cause gray mold disease in more than 1000 plant species, including many fruits and leafy vegetables, thereby causing huge annual economic losses (more than $10 billion) worldwide [1]. Due to its wide host range and economic impact, *B. cinerea* is considered as the second most agro-economically important plant pathogen after *Magnaporthe oryzae* [2]. *Botrytis cinerea* is generally considered as a necrotroph, which uses a diverse array of lytic enzymes and phytotoxins to damage or destroy host cells in order to acquire nutrients [1]. However, it has recently been shown that a short biotrophic phase (during 8–16 h post-inoculation) may play a crucial role in the pathogenesis of *B. cinerea* [3,4]. Hence, it may be more reasonable to classify *B. cinerea* as a hemibiotrophic phytopathogen [4] and to more thoroughly investigate the early infection stage of this pathogen to develop more effective management strategies.

Integrative approaches that incorporate good agronomic and horticultural practices are typically used along with chemical fungicides to control gray mold diseases [1]. Without the application of synthetic fungicides, crop losses to gray mold at pre- and post-harvest stages can reach 40% [5,6]. However, extensive uses of chemical fungicides have often been associated with severe consequences on human health and the ecosystem, urging the need for safer and eco-friendly alternatives [7,8,9]. Additionally, some fungicides (e.g., quinone outside inhibitor (QoI) class fungicides, such as azoxystrobin or trifloxystrobin) used against gray mold are no longer effective due to the emergence of resistant *B. cinerea* strains [10,11]. To overcome these issues, biotechnological approaches using genetic engineering have extensively been employed to control gray mold diseases. Host-induced gene silencing (HIGS) is one of the methods that have been used to enhance resistance against *B. cinerea*, by expressing dsRNAs that target essential fungal genes in host plant species leading to disease resistance [12,13]. For instance, *Arabidopsis* plants expressing hairpin RNA (hpRNA) targeting Dicer-like genes of *B. cinerea* [14] and transgenic potato and tomato plants expressing the dsRNA that is complementary to the target of the rapamycin (*TOR*) gene of *B. cinerea*, strongly reduced the gray mold occurrence in these host plants [15]. The HIGS can achieve durable resistance; however, due to the lack of a stable genetic transformations system for several economically important crops, the costs associated with development, registration, and maintenance of the genetically modified crops, as well as the challenges related to the public acceptance of these plants, utilization of HIGS as a disease management strategy against *B. cinerea* is not currently attainable.

Recently, the exogenous application of double-stranded RNA (dsRNA-), small interfering RNA (siRNA-), and hpRNA-mediated post-transcriptional gene silencing (i.e., RNA interference or gene silencing) has emerged as a promising strategy in enhancing plant resistance against phytopathogenic diseases [16]. For instance, topical applications of dsRNAs or siRNAs that target genes involved in the ergosterol biosynthesis in *Fusarium graminearum* (*CYP51A*, *CYP51B*, and *CYP51C*), suppressed the fungal growth in barley [17]. Similarly, spraying wheat plants with the dsRNA targeting *myosine 5* gene of *F. asiaticum* reduced fungal virulence [18]. In *Brassica napus*, exogenous applications of dsRNAs targeting various genes of *B. cinerea* also decreased the gray mold disease severity [19]. The dsRNAs or siRNAs targeting the *B. cinereal* siRNA biosynthesis-related genes, such as Dicer-like 1 and 2 (*DCL1* and *DCL2*), significantly reduced the gray mold diseases in various fruits and vegetables [14].

Crop losses due to *B. cinerea* during cultivation and post-harvest handling entail more specific and efficient management strategies against this phytopathogen. RNAi-based fungicides have attracted great interest for controlling fungal diseases, owing to their greater specificity and efficiency. This review aims to summarize recent observations and findings regarding the use of RNAi-based fungicides for controlling gray mold diseases, including the putative modes of action, applications, and limitations, especially in regard to the implementation of RNAi-based approaches for controlling gray mold diseases in the field.

## 2. RNAi: Biosynthesis and Actions

The biogenesis of siRNA is achieved by synthesizing dsRNA [20] from single-stranded RNA (ssRNA) templates using RNA-dependent RNA polymerase (RDR). RDR cleaves the dsRNA into small (mostly 20–30 nucleotides long) interfering RNA (siRNA) using Dicer endoribonucleases or Dicer-like (DCL) proteins [21], combining one strand of the siRNA (i.e., single-strand, or guide siRNA) and Argonaute (AGO) proteins to form RNA-induced silencing complexes (RISCs). The second strand (also known as the passenger strand) of the double-stranded siRNA is then degraded. The resulting RISC targets messenger RNA (mRNA) molecules that are complementary to the siRNA and degrades them, thereby inducing gene silencing (i.e., RNAi) [22].

The RNAi-based gene-silencing pathway has great diversifications in terms of its functionality. For instance, four homologs of DCL (DCL1, DCL2, DCL3, and DCL4) have been found in *Arabidopsis*; among these, DCL1 mainly cleaves the dsRNA and synthesizes the 18-21-nucleotides (nt), whereas DCL2, DCL3, and DCL4 synthesize the 22-, 24-, and 21-nt siRNA, respectively [23,24]. All these forms of siRNA play different roles in the plant. For instance, the siRNA 21-nt incorporates with AGO1 for the degradation of complementary mRNA sequences in a process collectively known as post-transcriptional gene silencing. However, when AGO1 is loaded with the siRNA 22-nt, it recruits RDR6 and transcribes it into dsRNA to activate secondary siRNA synthesis. The 24-nt siRNA, on the other hand, forms a complex with AGO4 and activates RNA-directed DNA methylation by using the DNA methyltransferase [25].

## 3. Small Interfering RNA (siRNA) Regulate the Virulence of *B. cinerea*

The world’s second most destructive phytopathogen, *B. cinerea*, employs a variety of virulence factors, including siRNA, to induce host susceptibility [21]. Such siRNA effectors play significant roles in the early stage of *B. cinerea* infection, as well as in the progression of gray mold diseases in a variety of crops and fruits [26]. *B. cinerea* uses siRNA effectors to manipulate the innate immunity of plant hosts by suppressing host defense responses (e.g., programmed cell death) to facilitate asymptomatic colonization (i.e., biotrophic phase), which occurs before the destructive necrotrophic phase. Previous studies have reported that *B. cinerea* DCL1 and DCL2 proteins synthesize and deliver siRNA to host plant cells to interfere with the host RNAi machinery of AGO1 and subsequently silences and suppresses the immune response of the host species [3]. The detailed characterization and profiling of the *B. cinerea* siRNA in infected plant tissue revealed that the *Bc*-siR3.2, *Bc*-siR5, and *Bc*-siR3.1 can target a variety of plant defense regulating factors, including genes for mitogen-activated protein kinases (e.g., *MPK1, MPK2*, and *MAPKKK4*), cell wall-associated kinase (e.g., *WAK*), and peroxiredoxin (e.g., *PRXIIF*), respectively [3] (Figure 1).

These genes play a central role in plant defense signaling pathways. For instance, mitogen-activated protein kinases (MPKs) are known to regulate hormone biosynthesis and signaling [27], thus activating the plant defenses against biotrophic [28], hemibiotrophic [29], and necrotrophic pathogens [30]. On the other hand, peroxiredoxin is one of the redoxins found in plant cells that regulates redox homeostasis (glutathione, ascorbate, NADPH-dependent redox), a process that has been linked to innate [31] and inducible immunity [32]. As far as the resistance to *B. cinerea* is concerned, the enhancement of glutathione- and ascorbate-dependent redox status is the prime defense responses in tomato against *B. cinerea* [32]. *Bc*-siRNA suppresses peroxiredoxin to manipulate the redox balance, hence increasing tissue susceptibility to *B. cinerea* infections [3]. Another virulence strategy that *B. cinerea* employs is the secretion of endopolygalacturonase that is involved in the degradation of host cell-wall pectins, releasing oligogalacturonides (OGs) [33]. The OGs are recognized as pathogen-associated molecular patterns (PAMPs) by the plant wall-associated kinase 1 (WAK1) and trigger a diverse array of plant defense responses, including enhanced camalexin biosynthesis [34,35]. To manipulate the cell wall-mediated defensive factors, *B. cinerea* produces siRNA that target WAK genes for silencing [14]. However, the detailed underlying mechanism of the RNAi-mediated regulations of cell wall-dependent defense remains largely unexplored.

Most of the predicted *Bc*-siRNA effectors are derived from retrotransposon regions [3]. However, Wang et al. [36] characterized an siRNA effector of *B. cinerea*, *Bc*-siR37 that is derived from a gene encoding a *Bc-ATPase* and that targets several immune responsive factors including the *WRKY7*, *WRKY57*, *PMR6*, and *FE12* in *Arabidopsis* (Figure 1). Among them, *FEI2* is a leucine-rich repeat (LRR) receptor kinase which is a plant immune-related transcriptional factor involved in effector-triggered immunity. *PMR6* encodes a pectinlyase and this, along with *FEI2*, are involved in the defense responses against *B. cinerea* [36]. On the other hand, the transcriptional factor *WRKY7* is known as a negative regulator of the salicylic acid (SA)-dependent *PR1* gene activation [37], and *WRKY57* is another transcriptional factor that binds to the JASMONATE ZIM-domain 4/8 (JAZ4/8) and activates the jasmonic acid (JA)-responsive defense genes [38]. Given the known role that JA plays in resistance against necrotrophic pathogens, along with the antagonistic relationships between SA- and JA-dependent defense pathways [31], it could be thus suggested that *B. cinereal* siRNA effector *Bc*-siR37 simultaneously silences the negative regulator of SA signaling to induce SA–JA antagonism and hence inactivates JA-mediated defenses against *B. cinerea*.

## 4. Plant siRNA Induce Immune Responses to Counteract *B. cinerea* Infections

siRNA and microRNA (miRNA) regulate a variety of cellular processes, including plant immunity [39]. RNA-dependent RNA polymerases (RDRs or RdRPs) are involved in the biosynthesis of dsRNA, which function as precursors for siRNA during siRNA synthesis [20]. Interestingly, *Arabidopsis* loss of function mutants *rdr6*, *dcl1-7*, and *dcl2/3/4* show greater susceptibility to gray mold [3,40], demonstrating the role of RNAi in defense against *B. cinerea*. On the other hand, Cai et al. [41] recently reported that *Arabidopsis* secretes extracellular vesicles, especially tetraspanin 8 (TET8)-associated exosomes that are loaded with plant siRNA (e.g., ta1c-siR483, ta2-siR453) in order to silence virulence genes in *B. cinerea* (Figure 2). Among fungal genes that are targeted for silencing are *vas51*, *dctn1*, and *sac1*, the three vesical trafficking genes associated with fungal pathogenicity [41]. Despite the significance of these results, the mechanisms underlying the loading of siRNA into vesicles and the transport of those vesicles into fungal cells remain largely unknown. More thorough research to understand the role of the vesical trafficking system in *B. cinerea*–host interactions could lead to the identification of important targets for RNAi-based fungicides and, more importantly, the discovery of effective methods for siRNA delivery to fungal cells.

Plant defense against invading pathogens relies on two forms of resistance—PAMP-triggered immunity (PTI) and effector-triggered immunity (ETI). Plant pathogens secrete effectors to suppress and interfere with both types of plant defenses. On the other hand, plants have evolved resistance genes (R-genes) and proteins (R-proteins) to directly or indirectly interact with and hinder the action of effector proteins, and subsequently activate the plant ETI responses [42]. Although most effectors are proteins, it was recently reported that *B. cinerea* employs siRNA effectors to suppress plant defenses via the host RNAi system [3]. The functional characterization of the RNAi-mediated regulation of hormonal signaling that regulates downstream defense responses [43] is necessary to fully understand in *B. cinerea*–host interactions.

## 5. RNAi-Based Biofungicides and Spray-Induced Gene Silencing (SIGS) for Controlling *B. cinerea*

Chemical fungicides are mostly used to control fungal diseases, including the gray mold diseases caused by *B. cinerea*. Fungicides with different modes of action are used against *B. cinerea*, including QoIs (e.g., anilinopyrimidines, dicarboximides, hydroxyanilides), succinate dehydrogenase inhibitors (e.g., boscalid, fluopyram), phenylpyrroles (e.g., fludioxonil), and hydroxyanilides (e.g., fenhexamid) [10,11,44,45]. Although synthetic fungicides offer an effective means of crop protection, there are many detrimental effects on human and plant health due to fungicide usage. The application of chemical fungicides impairs the plant’s metabolic and physiological pathways, such as photosynthesis, CO_2_ assimilation, reproductive organ development, and nitrogen, and carbon metabolism, and it may cause plant growth reduction and phytotoxicity [46,47]. Additionally, the extensive use of fungicides generates long-term residues in food and the environment. Some fungicides are also known to disrupt the human endocrine system and may cause abnormalities in the reproductive system [48]. A competitive alternative to synthetic fungicides would drastically reduce the indirect economic costs that fungicide usage inflicts on society. Moreover, multiple fungicide-resistant *Botrytis* strains have also been reported [10], especially in strawberry [49], raspberry [50], grape [51], and tomato [52]. Furthermore, the adverse effects of chemical fungicides on the environment and the lower efficacy of chemical fungicides against *B. cinerea* have been reported [1]. By reducing the use of synthetic fungicides and, therefore, the off-target exposure of fungicides to humans, non-target organisms, agricultural soils, and freshwater resources, billions of dollars can be saved each year. Therefore, new eco-friendly approaches should be implemented in the management of *B. cinerea*. The use of RNAi-based fungicides represents a promising alternative that could both control gray mold diseases and overcome the adverse effects of current strategies. Spray-induced gene silencing (SIGS) is an innovative RNAi-based approach for silencing target genes in phytopathogens using exogenous applications (Figure 3). The exogenous application of dsRNA and siRNA has been reported to reduce *B. cinerea* infection in strawberry and tomato fruits as well as detached leaves of oilseed rape [14,53], thereby initiating a new era of RNAi-based fungicide strategies for controlling gray mold diseases. Indeed, the application of siRNA and dsRNA targeting *B. cinereal DCL1/2* genes on the surface of fruits and vegetables alleviates gray mold diseases [14]. The *B. cinerea* double mutant *dcl1 dcl2* showed reduced virulence and disease progression in strawberry and tomato. However, the single mutant of the *dcl1 or dcl2* showed similar disease progression as the wild-type strain [14]. Another study reported that oilseed rape infection by *B. cinerea* was mitigated by topical applications of the dsRNAs that target *B. cinerea* genes, including *thioredoxin reductase*, *mitochondrial import inner membrane translocase subunit TIM44*, *peroxidase*, *pre-40S ribosomal particle*, and *necrosis- and ethylene-inducing peptide 2* [19]; on the other hand, dsRNAs targeting *thioredoxin reductase* and *mitochondrial import inner membrane translocase subunit TIM44* showed fewer necrotic lesions.

The virulence genes of the *B. cinerea* are a possible target for the RNAi-based fungicides, and this has been reviewed previously by Choquer et al. [53]. For instance, mutations in the *B. cinerea* chitin synthase genes (*Bcchs3a*) [54] or the genes involved in signal transduction including Gα subunits of G-proteins (*Bcg1*, *Bcg2,* and *Bcg3*) [55,56] reduce the virulence of *B. cinerea* in different plant species. However, more detailed transcriptomic analyses are still required to identify effective SIGS targets at different stages of *B. cinerea*–host interactions. Unlike HIGS, SIGS does not involve stable genetic transformation [16], making it a more acceptable alternative to the genetically modified organisms that require approval from various regulatory agencies. In addition, an SIGS-based disease management approach is considered a feasible option for combating fungicide-resistant *B. cinerea* strains. Indeed, dsRNA can be used to silence the regions of *B. cinerea* genes that are essential for fungicide resistance, rendering the existing fungicides more effective. Topical applications of dsRNA and siRNA are obstructed by issues associated with the stability of these molecules under open-field conditions where temperature, humidity, and consistent exposure to UV could considerably affect their stability. Typically, the efficacy of naked RNA molecules can only persist for a few days [57]. However, plants sprayed with dsRNA loaded onto nanoparticles (e.g., BioClay) have shown a greater degree of protection against pathogenic infections for 30 days after application, both on treated and newly emerged leaves [58]. Nanoparticles have also been reported to improve the uptake of the dsRNA by root tips and the silencing of the target genes [59]. To enhance the efficacy and stability of SIGS-based RNA biofungicides, the sustained release of RNA fungicides using nanoparticles as a carrier or stabilizer would be beneficial.

## 6. Conclusions and Future Perspectives

Currently, chemical fungicides are still the most common disease management strategy against invading fungal diseases. In the past few years, HIGS has emerged as a promising management strategy against pests, nematodes, and fungal pathogens in different crop species [60,61,62]. However, HIGS depends mainly on the genetic modification of host plants, and their applicability to horticultural crops remains controversial, not necessarily due to technical limitations, but mainly because of consumer concerns with transgenic crops. SIGS, on the other hand, does not require a stable genetic transformation of host species, predicting that their approval via the regulatory agencies would likely go through a fast track and their public acceptance would not be as concerning. SIGS could be efficiently used for the pre- and post-harvest management of gray mold diseases in major horticultural crops such as tomatoes, strawberries, and grapes. However, target genes for RNAi-based fungicides are still limited, thus the detailed transcriptomic analyses are warranted to identify target genes for biotrophic and necrotrophic phases of *B. cinerea* infections. Moreover, the uptake mechanisms of exogenous dsRNA by plants or fungal cells remain inexplicable. The clear understanding of the roles of membrane-bound proteins and receptors of plant and fungal cells may improve our understanding of the underlying small RNA uptake mechanisms. The stability of naked dsRNA molecules under field conditions is another major concern that may limit the application of SIGS-based disease management strategies. However, the utilization of nanoparticles and other stabilizers for enhanced stability and sustained release of the RNAi-biofungicides could overcome these limitations. The development of cost-effective and scalable approaches for the production of RNAi-biofungicides is probably the major challenge facing the practical agricultural utilization of this technology. However, technologies such as the bacterially expressed small RNA (dsRNA, and hpRNA) and minicells could solve these issues in the future. To conclude, RNAi-based biofungicides hold a great potential for managing the devastating gray mold diseases; however, future studies should focus on formulation, synthesis, stability, and application methods for the sustained release of RNAi-based fungicides, to make SIGS more effective, applicable, and cost-effective and to facilitate the implementation of SIGS under open-field conditions.

## Figures and Tables

**Figure 1 ijms-21-02072-f001:**
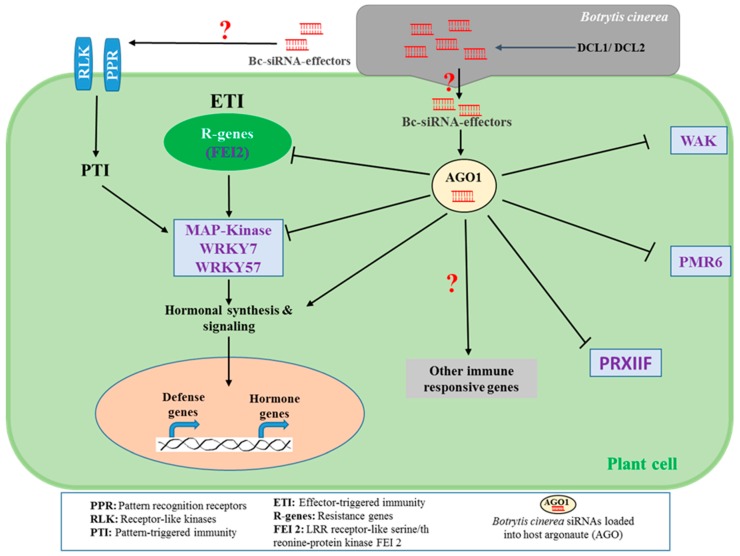
*Botrytis cinerea*’s small RNA (sRNA) effectors modulate the plant immune system to induce the gray mold disease. *B. cinerea* synthesize small interfering RNA (siRNA) effectors by Dicer-like 1 and 2 proteins (DCL1 and DCL2) and deliver them to host plant cells to interfere with the host RNAi machinery (argonaute; AGO1) and subsequently silence and suppress the plant defensive factors, including leucin rich repeat (LRR) receptor-like serine/threonine-protein kinase *FEI 2*, mitogen-activated protein kinases (*MAP-kinase*), *WRKY7*, *WRKY57*, cell wall-associated kinase (e.g., *WAK*), pectinlyase (e.g., *PMR6*), and peroxiredoxin (e.g., *PRXIIF*). The mechanisms by which *Bc*-siRNA interfere with the plant immunity systems (e.g., PTI and ETI), hormonal biosynthesis and signaling as well as immune responsive genes have been perceived but are not fully understood.

**Figure 2 ijms-21-02072-f002:**
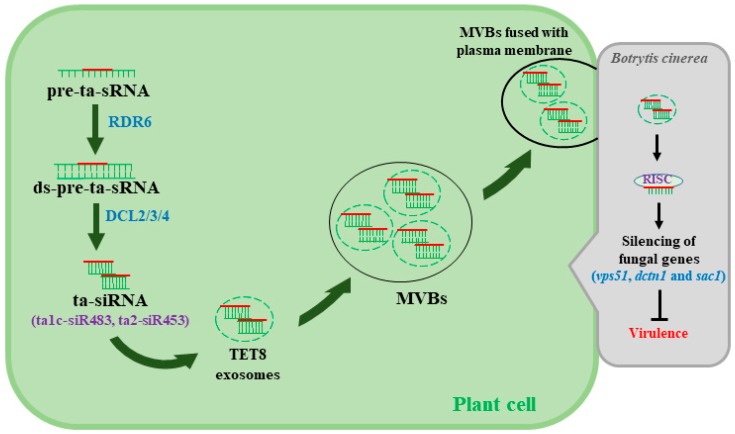
Plant delivers extracellular vesicles (EVs) containing siRNA to fungal cells to counteract *Botrytis cinerea*. Trans-acting siRNA (ta-siRNA) is processed by RNA-dependent RNA polymerase 6 (RDR6) and DCL2/3/4, and the processed ta-siRNA translocate through an unknown mechanism to tetraspanin 8 (TET8)-associated exosomes that are then assembled to form multivesicular bodies (MVBs). MVBs are fused with the plant plasma membrane to facilitate the transfer of siRNA to the invading fungal cells. The mechanisms behind the selection process of siRNA that translocate to the EVs and the release of siRNA into fungal cells to silence target virulence genes remain largely unknown.

**Figure 3 ijms-21-02072-f003:**
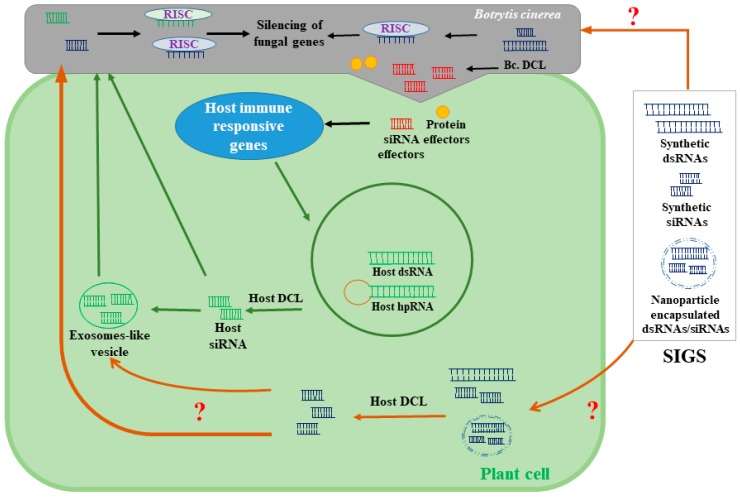
A model depicting possible mechanisms of spray-induced gene silencing (SIGS). SIGS using the exogenous applications of dsRNA or siRNA alleviates gray mold infections by targeting essential genes for *B. cinerea* virulence and pathogenicity. The mechanisms by which fungal cells uptake small RNA molecules are still unknown. It has also been proposed that sprayable forms of small RNA could be also transported to fungal cells through host cells. Although the uptake of exogenously applied siRNA and dsRNA by plant cells has already been shown, the mechanisms by which plants perceive and process exogenous RNA have not been identified. hpRNA: hairpin (hp) RNA; RISC: RNA-induced silencing complexes.

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
