# Peer review of "RNAi-Based Biofungicides as a Promising Next-Generation Strategy for Controlling Devastating Gray Mold Diseases"

_ijms, 2020, doi:10.3390/ijms21062072_

Round 1

Reviewer 1 Report

Botrytis cinerea is the second most agro-economically important plant pathogen and has been reported to cause gray mold diease in more than 1000 plant species. RNA-interference (RNAi)-based biofungicides are considered a promising strategy for controlling gray mold diseases. In this review, the small interfering RNAs (siRNAs) mediated regulation of host immunity and susceptibility in B. cinerea infections were summarized. And the efficiency and limitations of spry-induced gene silencing (SIGS) based RNAi-biofungicides were discussed.

It is a well-organized review. Some latest studies should be included.

Author Response

Reviewer 1:

Botrytis cinerea is the second most agro-economically important plant pathogen and has been reported to cause gray mold diease in more than 1000 plant species. RNA-interference (RNAi)-based biofungicides are considered a promising strategy for controlling gray mold diseases. In this review, the small interfering RNAs (siRNAs) mediated regulation of host immunity and susceptibility in B. cinerea infections were summarized. And the efficiency and limitations of spry-induced gene silencing (SIGS) based RNAi-biofungicides were discussed.

It is a well-organized review. Some latest studies should be included.

Response: We would like to thanks the reviewer for his/her positive evaluation on our manuscript. As per the reviewer’s suggestion the 15 new references and the relevant text have been added in the revised manuscript (see lines 39-66; 83-92; 240-251).

Reviewer 2 Report

Dear authors,

  1. around 15-25 new references should be added because it is review, 40 is not enough;
  2. information in the introduction is poor, please, expand;
  3. provide every section of your review with at least 1 figure;
  4. generally, the article is too small;
  5. it is better summarize answers than pose questions in the section  'Concluding remarks and future perspective';
  6. figures you provided might look better (personally I don't like question marks on figures.)

Author Response

Reviewer 2:

Response: We would like to thanks the reviewer for allowing us to revise the manuscript. All comments made by the reviewers were carefully addressed. In the revised version of the manuscript, we elaborated in some sections to provide extra information and recent literature. We have also made significant revisions to the manuscript to improve clarity and readability. We are hoping that the reviewer finds the current version of the manuscript acceptable and publishable.

Q1. Around 15-25 new references should be added because it is review, 40 is not enough;

Response: As per reviewer’s suggestion, 15 new references with the relevant text are added in the revised manuscript (see lines 39-66; 83-92; 241-252).

Q2. Information in the introduction is poor, please, expand;

Response: According to reviewer’s suggestion, the introduction section has been elaborated in the revised text (see lines 39-66).

Q3. Provide every section of your review with at least 1 figure;

Response: We have initially suggested and mainly focus on the RNAi-based biofungicides for B. cinerea in this current review. The literature of the RNAi-biofungicides against the gray mold diseases is limited. Authors believe both figures are relevant to summarize the available information in the literature regarding the RNAi-biofungicides against the gray-mold disease.

Q4. Generally, the article is too small;

Response: The idea of this manuscript is to summarize the available information and future possibilities of the RNAi-based next-generation biofungicides to control the gray mold diseases in various crops. To the best of our knowledge, the available information regarding the RNAi-biofungicides against B. cinerea has been discussed in this manuscript. As per the reviewer’s suggestion, we elaborated in some sections including the introduction (see lines 39-66), RNAi biosynthesis and action (see lines 83-92), and the future perspective (see lines 241-252) sections in the revised manuscript to provide extra information and latest literature.

Q5. It is better summarize answers than pose questions in the section 'Concluding remarks and future perspective';

Response: According to reviewer suggestions, the authors describe the possible answers and revise the concluding remarks and future perspective section in the revised text (see lines 241-252).

Q6. Figures you provided might look better (personally I don't like question marks on figures.)

Response: RNAi-mediated B. cinerea-host interactions are poorly known and the mode of actions of RNAi-biofungicides including the uptake of the exogenous applied small RNA (dsRNA, hpRNA, siRNA) by plant and/or fungal cells, RNAi mediated regulations in plant immunity system remains in dark. By giving the quotation marks it would be much informative for the reader and beneficial for future research to make RNAi-biofungicides available for practical application in crop protection.

Round 2

Reviewer 2 Report

Dear authors, 

please, add some more figures to your review:

a) RNAi: Biosynthesis and actions including peculiar features known for fungi and B. cinerea particularly;

b) how siRNAs induce immune responses to counteract B. cinerea infections.

Please, expand,

"Although synthetic pesticides offer an effective means of crop..."

a) what synthetic pesticides are used against B. cinerea?

b) what problems meet synthetic insecticides in controlling B. cinerea?

Provide at least 7-10 references while answering these questions.

Author Response

We would like to thank you for his/her positive comments on our work and for allowing us to revise the manuscript to improve the quality. The manuscript has now revised according to reviewers’ comments. All comments made by the reviewers were carefully addressed. The corrections or changes were indicated in the response sheet and all the revised sections were marked in red color.

Q.1) please, add some more figures to your review:

Response: According to the reviewer's suggestion, the authors added one more additional figure regarding host siRNA induced defense responses against B. cinerea in the revised manuscript (see Figure 2).

Q.2) a) RNAi: Biosynthesis and actions including peculiar features known for fungi and B. cinerea particularly;

Response: The addition of a figure regarding RNAi biosynthesis in fungi would indeed be beneficial. However, RNAi biosynthesis and actions, especially in fungi, have been reviewed in recent reviews (Majumdar et al., 2017 and Machado et al., 2018) and it will be a sort of unjustified repetition to include this in our review.

  • Majumdar et al., 2017; RNA Interference (RNAi) as a Potential Tool for Control of Mycotoxin Contamination in Crop Plants: Concepts and Considerations, Front Plant Sci. 2017; 8: 200.
  • Machado, A.K.; Brown, N.A.; Urban, M.; Kanyuka, K.; Hammond-Kosack, K.E. RNAi as an emerging approach to control Fusarium head blight disease and mycotoxin contamination in cereals. Pest Manag. Sci. 2018, 74, 790–799.

Q.3) b) how siRNAs induce immune responses to counteract B. cinerea infections.

Response: The authors added one more additional figure explaining how host siRNAs may induce immune responses to counteract B. cinerea infections. We have also explained this in the revised text (see lines 171-177; Figure 2, in the revised text).

Q.4) Please, expand,

"Although synthetic pesticides offer an effective means of crop..."

  1. a) what synthetic pesticides are used against B. cinerea?
  2. b) what problems meet synthetic insecticides in controlling B. cinerea?

Provide at least 7-10 references while answering these questions

Response: As per the reviewer’s suggestions, authors added five more new relevant references and provided additional information regarding the chemical fungicides that are used for management of gray mold diseases (see lines 180-183 in the revised text) and the consequences associated with fungicide usage (see lines 185-190 in the revised text).